# The Influence of Optical Alignment Error on Compression Coding Superresolution Imaging

**DOI:** 10.3390/s22072717

**Published:** 2022-04-01

**Authors:** Chao Wang, Siyuan Xing, Miao Xu, Haodong Shi, Xingkai Wu, Qiang Fu, Huilin Jiang

**Affiliations:** 1Jilin Provincial Key Laboratory of Space Optoelectronic Technology, Changchun University of Science and Technology, Changchun 130022, China; xsyjyy@163.com (S.X.); xumiao89@126.com (M.X.); shihaodong08@163.com (H.S.); wk_xkg@163.com (X.W.); strich@sina.com (Q.F.); jhl20092014@cust.edu.cn (H.J.); 2College of Optoelectronic Engineering, Changchun University of Science and Technology, Changchun 130022, China

**Keywords:** alignment tolerance, compression coding, decenter and tilt, optical design, Superresolution imaging

## Abstract

Superresolution (SR) imaging technology based on compression coding has always been considered as the key to break through the geometric resolution of the detector. In addition to factors such as the reconstruction algorithm and mounting platform vibrations, the impact of inherent errors in the optical system itself on the reconstruction results of SR imaging is also obvious. To address this issue, a study on the design of the SR optical system and the influence of optical alignment errors on SR imaging was conducted. The design of the SR optical system based on digital micro-mirror device (DMD) for long-wave infrared wavelength was completed, and an athermal analysis of the system was carried out. The design results showed that the SR optical system has good imaging quality in the operating temperature range. The imaging model of the DMD SR imaging optical system is established according to the designed SR optical system. We investigated the influence of various alignment errors, including decenter, tilt, lens interval error and defocus, on the imaging properties of the SR optical system. Various random combinations of alignment errors were introduced into the optical system, respectively, and the SR reconstructed image quality of the imaging system was analyzed using the inverse sensitivity method to obtain the tolerance limits when the system was assembled. Finally, the effectiveness of the method to obtain the alignment tolerance limit of the compression coding SR imaging optical system was verified through a desktop demonstration experiment.

## 1. Introduction 

Superresolution (SR) imaging is a class of techniques that enhance the resolution of an imaging system. In optical SR the diffraction limit of systems is transcended, while in geometrical SR the resolution of digital imaging sensors is enhanced [1]. Among many SR imaging methods, the compressed sensing (CS) method based on the principle of signal sparsity has attracted much research attention in recent years [2,3]. Based on the compressed sensing theory and the aperture coding device, which can be abbreviated as ‘compression coding’, the SR image system manages to improve the resolution in all regions of the image, possessing good imaging quality and breaking the limitation of the image sensors’ size without increasing cost constraint of the large-array sensors [4].

CS theory seeks to minimize the amount of input data, and this concept is similar to the concept of reconstructing a high-resolution image from low-resolution images in SR imaging, where the low-resolution images can be understood as “compressed” data [5]. Since CS theory was originally used in the field of signal processing and computer science, most of the current research on compression coding SR imaging technology is focused on the reconstruction algorithms based on CS theory. Yang et al. improved the effect and performance of CS in SR imaging by introducing a sparse training dictionary [6]. Sun et al. improved the reconstruction results of compression coding SR imaging based on redundant dictionary is not satisfactory in terms of noise suppression and edge sharpness [7]. Zhang et al. proposed an optional frequency–domain compressive coding technique, which solved the problem of the high computational cost of compressive-sensing-based SR algorithms in the spatial domain [8]. Zhou et al. presented a new SR imaging approach based on compression coding and sequence information which had better performance in both subjective assessment and objective standards, including entropy and average gradient [9]. Zhang et al. proposed a novel remote sensing system that uses high-resolution MEMS devices to achieve a resolution that exceeds that of image sensors, and by applying the theory of compressed sensing, the rate of data collection is compressed while maintaining high image quality, reducing the cost of high-resolution images with the sensor [4]. The above research has achieved good SR imaging results, but the influence of the optical system itself on the compression coding SR reconstruction results has been ignored. In the existing research related to the optical path factors in compression coding imaging, Dumas, P. improved the reconstruction model to address the problem that light from individual mask elements can leak onto sensor pixels neighboring the geometrically imaged pixel during experiments [10]. Zhu proposed an adjustment scheme for a coded aperture imaging spectrometer system. Aiming at the problem of optical axis misalignment caused by multiple reflections of the optical path, a method combining laser reference positioning and light refraction and reflection principle was proposed for alignment [11]. In fact, various errors including decenter, tilt, lens interval error, defocusing, etc. will be introduced in the optical system during the installation and alignment process. If the impact of each error on the compression coding SR optical imaging could be quantified, the alignment process of this SR imaging system can be guided more effectively, and the alignment tolerance list can be given accurately.

In this paper, the imaging model of the DMD SR imaging optical system was firstly established, and a DMD SR imaging optical system in the long-wave infrared band was athermally designed. The compression coding SR imaging simulation scheme with ZEMAX optical software was proposed, and the analysis method of the influence of the adjustment error on the SR imaging quality was presented. All kinds of alignment errors were introduced into the SR imaging model, and the tolerance range of the DMD SR optical imaging system was obtained.

## 2. Imaging Model and Simulation of Compression Coding SR Imaging Optical System

Blurred images can often be represented as nonstationary 2D stochastic processes that can be modeled by a set of linear space-varying state equations. The degraded image can be expressed as the convolution of the original clear image and the blur kernel function:(1)b(x,y)=i(x,y)∗k(x,y)+(x,y)
where, (*x*, *y*) represents the coordinates of the blurred image, *b*(*x*, *y*) represents the blurred image, *i*(*x*, *y*) represents the original clear image, *k*(*x*, *y*) represents the blur kernel function, and *n*(*x*, *y*) represents the additive noise [12].

For the optical image degradation process, the blur kernel function *k*(*x*, *y*) causing image blur is the point spread function (PSF) of the optical system. Generally, the optical blur degradation process can be regarded as a linear spatial variation degradation process, and the blurred image can be expressed as:(2)$$b\left(x,y\right)=\int_{-\infty }^{+\infty }\int_{-\infty }^{+\infty }i\left(s,t\right)k_{s,t}\left(x-s,y-t\right)d_{s}d_{t}+n\left(x,y\right)$$
where (*s*, *t*) represents the coordinates of the original image, and *k_s,t_* represents the spatially varying PSF [13].

The convolution of the original image and the PSF of the optical system is used to simulate the imaging process of the optical system, and the influence of noise is ignored in the simulation. The schematic diagram of the optical system used in the simulation is shown in Figure 1a, and the simulation flow chart is as Figure 1b:

The simulation process was then realized by the MATLAB software. The simulation mainly includes the following contents:(1)The scene is convolved with the PSF of the telescopic objective to simulate the imaging process of the telescopic objective.

Since the imaging quality of the telescopic objective differs at different field of view (FOV) point, we set up 3 × 3 uniformly distributed FOV points and used ZEMAX to simulate the telescopic objective to obtain the PSF matrixes of these 9 FOV points. The 3 × 3 FOV points setting is shown in Figure 2. When PSF is acquired through ZEMAX, it should be ensured that the sampling interval of PSF data is the same as the pixel size of the detector.

According to the extracted PSF of the 9 sample points, the PSF of other FOV points are obtained by interpolation to obtain the influence of the telescopic objective lens on the entire FOV. The method used is the PSF interpolation method based on principal component analysis (PCA). The basic idea of this method is to find a set of basis functions, and the known PSF data of the imaging system can find a good approximation in the function space composed of this set of basis functions. The PSF acquisition method based on PCA includes 3 steps: (1) Characterization of PSF of the known points: Utilize PCA model to characterize the PSF of the known points, and obtain the PSF principal component analysis model with these known points; (2) PSF model parameter interpolation calculation: Use point interpolation method to obtain the parameters of PSF model of the unknown points; (3) Regression of PSF of the unknown points: Bring the acquired parameters of the unknown point PSF into the PSF representation model to obtain the PSF of the unknown points [14]. At this time, we obtain the PSF distribution of the full field of view, which is convoluted with the scene to obtain the image of the scene on the DMD through the telescopic objective.
(2)The obtained image in (1) is modulated by superimposing the image with the DMD coding template. An SR imaging method based on block compressed sensing is used.

The 4 × 4 pixels of the DMD are considered as a ‘block’, which is used to correspond to 1 pixel of the long-wave infrared (LWIR) focal plane array (FPA). Each block is reconstructed separately, and the image resolution is 4 times that of FPA. The coding used is a random Bernoulli matrix. A 4 × 4 random Bernoulli matrix is generated, which is then cyclically tiled to a 4a × 4b matrix and dotted with the convolved image obtained in (1);
(3)The encoded image is convolved with the PSF of the projection objective to simulate the imaging process of the projection objective. The PSF acquisition method of the projection objective is the same as the method in (1);(4)Downsampling the obtained 4a × 4b image in (3) to a 1 a × b image to simulate the imaging process of a low-resolution LWIR FPA;(5)12 images with a resolution of a × b are reconstructed into one image of 4a × 4b by the recovery algorithm, so that the image reconstruction with 4 times resolution is realized. Therefore, the compression ratio is 12/16 = 75%.

In the SR reconstruction, the same reconstruction operation is performed for each block. The reconstruction algorithm adopted is the Orthogonal Matching Pursuit (OMP) [15]. In this algorithm, the approximations derived from a matching pursuit can be refined by orthogonalizing the directions of projection. The orthogonalized matching pursuit algorithm converges in a finite number of steps in finite-dimensional spaces [16], and the detailed algorithmic procedure can be found in [17]. We use the discrete cosine transform (DCT) as the basis for the reconstruction [18]. Because the block size used is 4 × 4, the DCT matrix size is 16 × 16. Then, the image blocks with resolution of 4 × 4 are reconstructed by using the DCT matrix, the coding matrix and 12 gray values obtained by one pixel in the detector, and the blocks are connected together to obtain a complete high-resolution image. The peak signal to noise ratio (*PSNR*) value is used as the objective evaluation index of the image [19]. The unit of *PSNR* is dB. The larger the *PSNR* value, the smaller the distortion. When the *PSNR* is lower than 20 dB, the image restoration effect is poor. *PSNR* is most easily defined via the mean squared error (*MSE*). Given a noise-free m × n monochrome image I and its noisy approximation *K*, *MSE* is defined as
(3)$$MSE=\frac{1}{mn}\sum_{i=1}^{m-1}\sum_{j=1}^{n-1}{\left[I\left(i,j\right)-K\left(i,j\right)\right]}^{2}$$

The *PSNR* (in dB) is defined as
(4)$$\begin{array}{ll} PSNR & =10\cdot \log_{10}\left(\frac{MAX_{I}^{2}}{MSE}\right)=20\cdot \log_{10}\left(\frac{MAX_{I}^{2}}{\sqrt{MSE}}\right)\\ & =20\cdot \log_{10}\left(MAX_{I}\right)-10\cdot \log_{10}\left(MSE\right) \end{array}$$
here, *MAX**_I_* is the maximum possible pixel value of the image. In this paper the pixels are represented using 8 bits per sample, and *MAX**_I_* is 255. More generally, when samples are represented using linear PCM with B bits per sample, *MAX**_I_* = 2*^B^* − 1.

## 3. DMD-Based SR Imaging Optical System Design

(1)Structure of DMD-based SR optical system.

According to the above-mentioned basic imaging model of the SR imaging optical system, a DMD-based SR imaging optical system is designed. DMD is essentially a reflective digital semiconductor light modulator, which is composed of millions of micro mirrors on a semiconductor silicon substrate [20]. Each micro mirror is independently controlled by static electricity to determine the light deflection angle [21].

The DMD is placed on the optical intermediate imaging surface of the system as a coded aperture mask. The front coding element and the back data processing cooperate to form a hybrid optical-digital computing imaging system.

The imaging optical system is mainly composed of five parts: target source, telescopic objective lens, DMD, projection objective lens, LWIR FPA. The target scene is imaged on the middle image plane by the telescopic objective lens. Then the middle image is modulated by the DMD and finally imaged on the FPA through the projection objective lens. The overall composition of the imaging system is shown in Figure 1a, and the main system parameters are shown in Table 1:

In the optimization process, the image on the DMD is tilted relative to the folded optical axis because the DMD deflects the light by 24°, so it is necessary to set the FPA imaging tilt as a variable for optimization [22]. At the same time, in order to avoid interference between the last lens of the telescopic objective and the light reflected by the DMD, the back-working distance of the telescopic objective and the distance from the DMD to the projection objective need to be controlled [23]. The structure diagram of the optical system is shown in Figure 3, and the modulation transfer function (MTF) of the optical system is shown in Figure 4. It can be seen that the optical system has good imaging quality, and the MTF from 0 to the FPA cutoff frequency is close to the diffraction limit.

The PSF of the telescopic objective and projection objective in this design were brought into the imaging framework of the compression coding SR imaging system in Section 2, and the obtained low-resolution images are reconstructed by OMP algorithm. Finally, the SR reconstructed image is shown in Figure 5b, with a *PSNR* value of 22.9670.
(2)Athermal design.

For the long wave infrared imaging system, the temperature variation has a strong influence on the image quality, so the athermal design is highly necessary for widening the applicable temperature range of the system. There are two methods of athermalization: active and passive. Passive athermalization eliminates the thermal focus shift of the system by properly combining optical materials. This approach eliminates the need for moving parts in the system, increases the overall reliability of the system, and the optical and mechanical structure of the system can be very simple, compact, and lightweight. Therefore, we chose the optical passive athermalization method. With this method, the performance parameters of selected optical materials and the structural parameters of the system should meet the following equations:

Optical power equation: (5)$$\sum_{i=1}^{k}h_{i}\phi_{i}=\phi$$

Achromatic equation: (6)$$C=\sum_{i=1}^{k}\frac{h_{i}^{2}C_{i}\phi_{i}}{\phi }=0$$

Athermal aberration equation: (7)$$L=\sum_{i=1}^{k}\frac{h_{i}^{2}C_{i}\phi_{i}}{\phi }=\alpha_{m}L$$
where *k* is the total number of optical surfaces, *h_i_* is the distance from the incident point of the paraxial light on each lens to the optical axis, *C_i_* is the chromatic aberration coefficient, $\phi_{i}$ is the optical power of each lens, $\alpha_{i}$ is the linear expansion coefficient of the lens tube material, *L* is the total length of the system, and *T_i_* is the coefficient of thermal aberration of the lens material.

Since the temperature variation range of the environment where the system is located is −10 °C–40 °C, this system is athermally designed for this temperature range. The MTF of the system at different temperatures can be obtained directly through ZEMAX. Therefore, if the MTF value is too low in the required temperature range, lens optical parameters such as radius of curvature, glass thickness and material need to be adjusted, and the overall structural parameters of the system also need to be optimized in order to achieve the desired image quality. Figure 6 shows the MTF curves of the DMD SR imaging optical system at −10 °C, 10 °C, 20 °C, and 40 °C after the athermal design, and Table 2 shows the *PSNR* and the structural similarity index (SSIM) values of the reconstructed images of the optical system at the above temperatures. It can be seen that the imaging quality of the system is well maintained in the operating temperature range. 

## 4. Analysis of the Influence of Lens Alignment Error on SR Reconstruction

The alignment errors including decenter, tilt, air interval error and defocus are introduced respectively into the SR imaging optical system designed in this section. The optical system’s PSF on typical FOV points when the alignment error is introduced into the system can be obtained from ZEMAX. Then the full FOV PSF data can be obtained through the interpolation method in Section 2 and brought into the based-on DMD SR imaging model.

The FOV angle of the SR imaging optical system designed in this paper is different in the X and Y directions. When introducing the adjustment error into the optical system, it is necessary to analyze the errors that can be tolerated in the X and Y directions for each lens separately.

There are three main alignment tolerance analysis methods: sensitivity analysis, inversion sensitivity analysis and Monte Carlo analysis [24]. The sensitivity and inverse sensitivity analysis considers the effects on system performance for each tolerance individually. The aggregate performance is estimated by a root-sum-square calculation. The Monte Carlo analysis generates a series of random lenses which meets the specified tolerances, then evaluates the criterion. No approximations are made other than the range and magnitude of defects considered [25]. In the paper, the inverse sensitivity analysis method was used to assess the system performance with the *PSNR* decline value of the reconstructed image not exceeding 1 as the evaluation criterion for the tolerance analysis. 

### 4.1. Influence of Lens Decenter on SR Reconstruction Results

A gradually increasing decenter error is introduced for each lens of the optical system in X and Y directions in steps of 0.01 mm. The schematic of the XOY coordinate system can be seen in Figure 3. The PSF with the decenter error is substituted into the imaging model and the image reconstruction is performed. Figure 7 shows the relationship between the decenter of each lens and the *PSNR* value of the reconstructed image. The correspondence between the lenses and the serial numbers in Figure 8 is shown in Figure 4.

It can be seen from Figure 7 that the decenter error of the seventh lens has the greatest impact on the reconstruction results. When 0.08 mm de-center error is introduced in the X direction or 0.06 mm error is introduced in the Y direction for the seventh lens, the *PSNR* value of the reconstructed image decreases by 1.

### 4.2. Influence of Lens Tilt on SR Reconstruction Results

A gradually increasing tilt error is introduced for each lens of the optical system in X and Y directions in steps of 0.01°. Then the reconstructed image with lenses tilt is obtained according to the previous SR imaging model. Figure 8 shows the relationship between the tilt of each lens and the *PSNR* value of the reconstructed image.

From Figure 8, it can be seen that the *PSNR* value of the reconstructed image decreases by 1 when the second lens is tilted 0.13° in the X direction or 0.08° in the Y direction of the sixth lens. 

### 4.3. Influence of Lens Interval Error on SR Reconstruction Results

The distance between each two lenses is changed, increasing by 0.01 mm each time, and then the image is reconstructed using the previous imaging model. Figure 9 shows the relationship between the lens interval error and the *PSNR* value of reconstructed image.

Since the object distance of the SR optical system is at infinity, the interval error between the first lens and the object plane has no impact on the reconstruction result. The sixth interval error, that is, the spacing error of the fifth and sixth lens has the greatest impact on the image reconstruction result. When the sixth interval error is 0.06 mm, the *PSNR* value of the reconstructed image decreases by 1.

### 4.4. Influence of Optical Defocus on SR Reconstruction Results

The distance between the telescopic objective and the intermediate image plane (Distance 1), and between the projection objective and the image plane (Distance 2) are changed, respectively. These distances, i.e., the defocus is gradually increased in steps of 0.01 mm, and then reconstructed image is obtained using the above imaging model. Figure 10 shows the relationship between the defocus of the optical system and the *PSNR* value of the reconstructed image.

It can be seen in Figure 10 that the defocus of the telescopic objective and the projection objective has a great impact on the reconstruction results. When the intermediate image plane is defocused by 0.05 mm or the image plane is defocused by 0.04 mm, the *PSNR* value of the reconstructed image decreases by 1.

Finally, a combination of the above four errors is introduced into the system simultaneously and 100 alignment simulations are performed. In each alignment, we control the maximum value of each error not to exceed the value that makes *PSNR* decrease by 1 when it is introduced separately. The inverse sensitivity method is employed to analyze the tolerance of the SR imaging system. When multiple errors are introduced simultaneously, the results of the tolerance assignments for a DMD super-resolution imaging optical system with a *PSNR* drop of no more than 1, which is shown in Table 3.

## 5. Imaging Experiments and Results Discussions

### 5.1. Experimental Device and Process

In order to verify the accuracy of the proposed DMD SR imaging optical system setup error analysis method, a DMD SR imaging experimental platform was built to perform SR reconstruction of the collected target scene. The schematic diagram of the experiment is shown in Figure 11.

Due to the limitation of the experimental conditions, we use the DMD and CCD camera in visible band to verify the optical system alignment error analysis method proposed above. In the visible band experimental optical path, the lens parameters used are known. The optical path is simulated in ZEMAX. The MTF of the system without alignment error is shown in Figure 12. Using the optical system PSF acquired by ZEMAX, the imaging model of this DMD-based SR imaging optical system is constructed according to the method in Section 2. Appropriate decenter, tilt and defocus errors are introduced into the model in turn, and the simulation curve of the *PSNR* value of the reconstructed image with the error is obtained.

The experimental procedure is briefly described below. Due to the limitations of the experimental conditions, we use the DMD and CCD camera in visible band to verify the optical system alignment error analysis method proposed above. The actual experimental setup is shown in Figure 13. The scene is imaged on the DMD by the fixed-focus lens and the bi-telecentric objective lens 1. The scene is encoded and modulated by controlling the reflection direction of the different micromirrors of the DMD. The encoded scene image is imaged on the detector through the bi-telecentric objective lens 2. The 4 × 4 pixels of the detector are combined into one pixel to simulate the downsampling process of low-resolution infrared FPA. Then, the SR reconstruction of the obtained encoded image is carried out, and the *PSNR* value of the reconstructed image is calculated by taking the image directly collected without binning pixels and encoding as the original reference image (ground truth). By changing the position of the bi-telecentric objective lens 2 in the optical path, the adjustment errors of the same size and direction as in the simulation of the previous paragraph are sequentially introduced, and the coded images are acquired again for reconstruction, and the *PSNR* value of the reconstructed images with the alignment error is introduced is calculated to obtain the corresponding curve between each error and *PSNR* value of the reconstructed image.

The experimental results are compared with the simulation results to verify the validity of the proposed model of the influence of the alignment error on the SR imaging quality.

### 5.2. Results and Discussions

The *PSNR* value of the ideal reconstructed image obtained by the simulation is 32.8601, and the corresponding curve between each adjustment error and the *PSNR* value of the reconstructed image is shown in Figure 14a.

In the desktop demonstration experiment, the image with DMD fully lit and without pixel binning, the high-resolution image reconstructed by coded low-resolution image without alignment error and the reconstructed high-resolution image with various errors introduced are shown in Figure 14. The *PSNR* of the experimental reconstructed image without error is 25.2547. In the simulation and experiment, the *PSNR* of the reconstructed image varies with the alignment error as shown in Figure 15.

In the actual experiments, there are certain pixel alignment errors and lens aberrations in the optical path which will have some influence on the results of SR reconstruction, making the *PSNR* value of the experimentally obtained reconstructed image lower than the *PSNR* value of the simulated reconstructed image. However, according to the relationship curves between the *PSNR* value obtained by simulation and experiment and the alignment error, it can be seen that the variation trend of *PSNR* value with the introduced alignment error in the simulation is basically the same as that in experiment, which verifies the effectiveness of the proposed method for analyzing the alignment tolerance of the compression coding SR optical systems.

## 6. Conclusions

The PSF of each part of the DMD-based compression coding SR imaging optical system was obtained by the optical design software, and the imaging model of this system was established. A DMD-based SR LWIR imaging optical system was designed, and an athermal design for it was carried out. For this optical system, an analysis method of the influence of alignment error on the SR imaging quality was proposed. The allowable alignment tolerance range of this LWIR SR imaging optical system was determined, that is, the de-center error of each lens in the optical system in X and Y directions should be controlled within ±0.06 mm and ±0.05 mm, respectively; the tilt error of each lens in X and Y direction should be controlled within ±0.05°; the error of the interval between two lens should be controlled within ±0.02 mm; the defocusing amount of the intermediate image plane should be controlled within ±0.04 mm; the defocusing amount of the final image plane should be controlled within ±0.03 mm. Through the simulation and the desktop demonstration experiment, the effectiveness of the analysis method proposed in this paper for the influence of the alignment error on the SR reconstruction quality was verified. In the next step, by accurately designing the magnification of the infrared projection objective and strictly controlling the alignment error of the lens and the system according to the tolerance table given in Section 4 of this paper, the SR imaging with pixel-wise registration will be realized. 

## Figures and Tables

**Figure 1 sensors-22-02717-f001:**
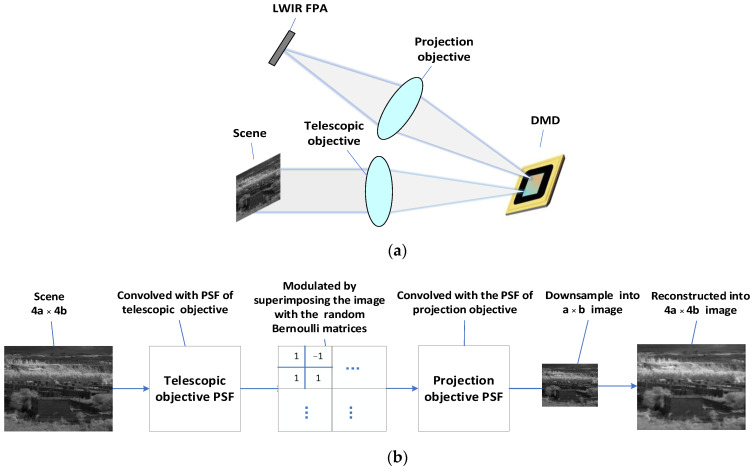
Simulation flowchart: (**a**) schematic diagram of optical system; (**b**) simulation flowchart model.

**Figure 2 sensors-22-02717-f002:**
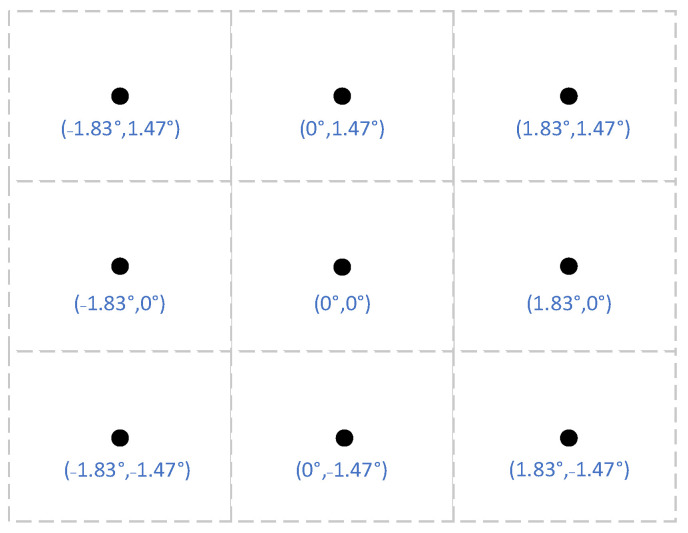
3 × 3 FOV points settings.

**Figure 3 sensors-22-02717-f003:**
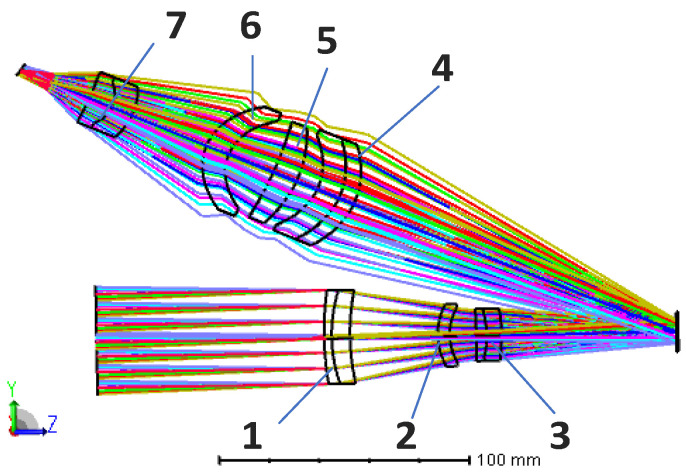
Structure diagram of optical system.

**Figure 4 sensors-22-02717-f004:**
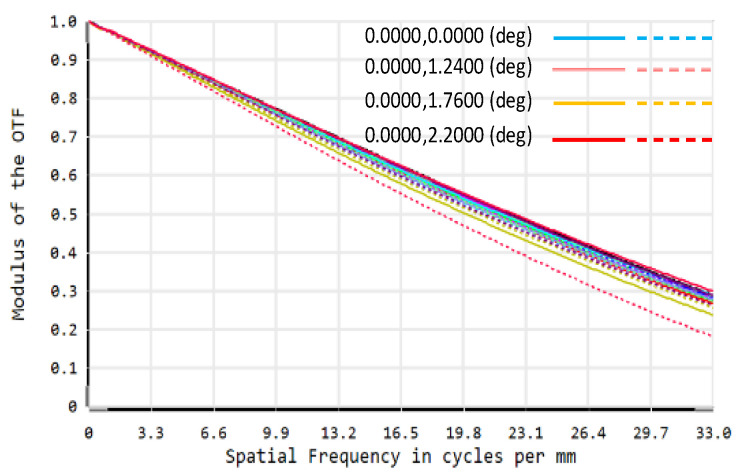
Modulation Transfer Function (MTF) of optical system.

**Figure 5 sensors-22-02717-f005:**
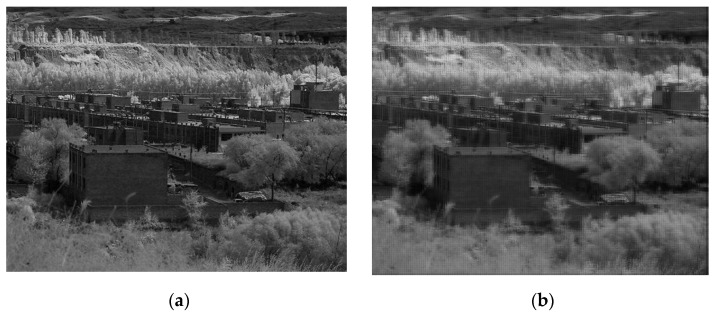
Original image and OMP reconstructed image: (**a**) original image; (**b**) image degenerated by PSF and reconstructed by OMP.

**Figure 6 sensors-22-02717-f006:**
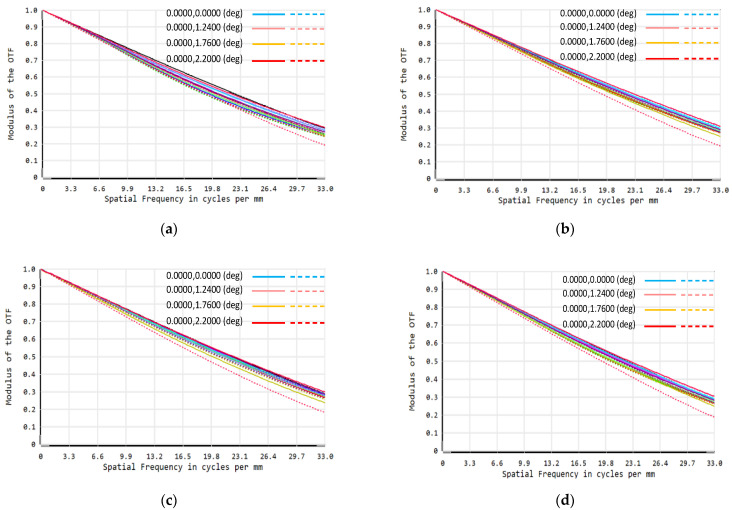
DMD SR optical system MTF at different temperatures: (**a**) −10 °C; (**b**) 10 °C; (**c**) 20 °C; and (**d**) 40 °C.

**Figure 7 sensors-22-02717-f007:**
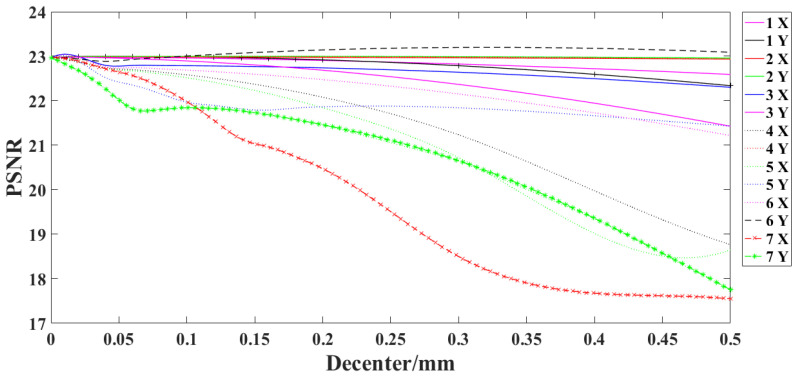
Relationship between lens decenter and *PSNR* value of reconstructed image.

**Figure 8 sensors-22-02717-f008:**
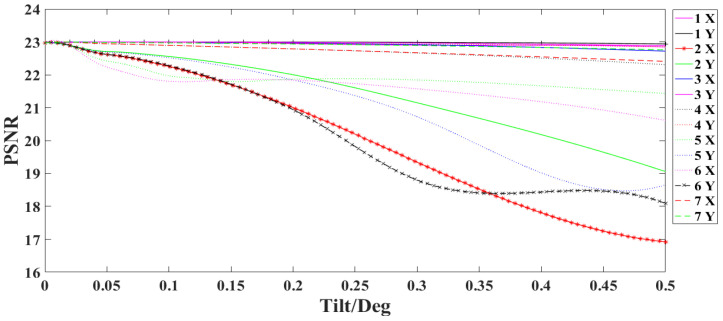
Relationship between lens tilt and *PSNR* of reconstructed images.

**Figure 9 sensors-22-02717-f009:**
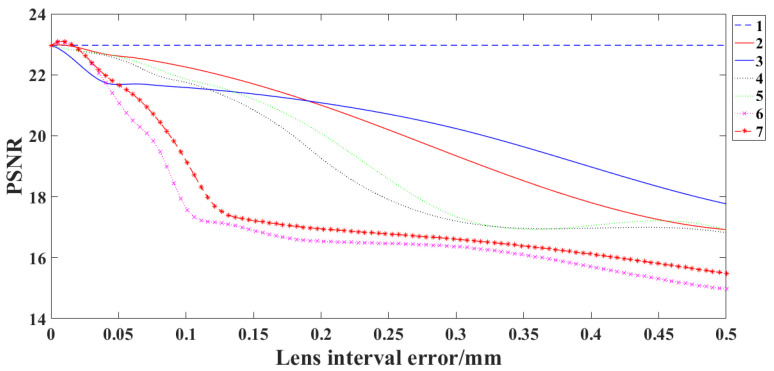
Relationship between lens interval error and *PSNR* of reconstructed images.

**Figure 10 sensors-22-02717-f010:**
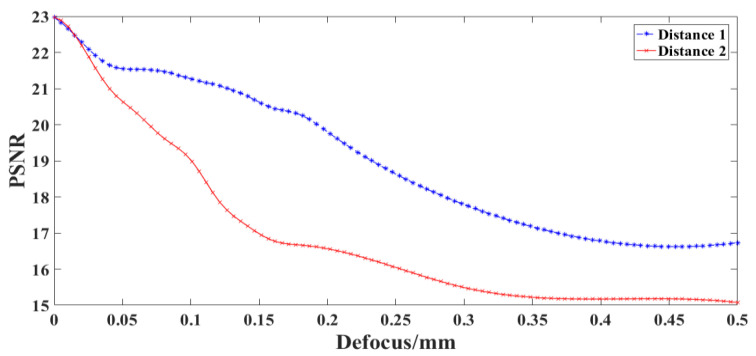
Relationship between defocus of optical system and *PSNR* value of reconstructed image.

**Figure 11 sensors-22-02717-f011:**
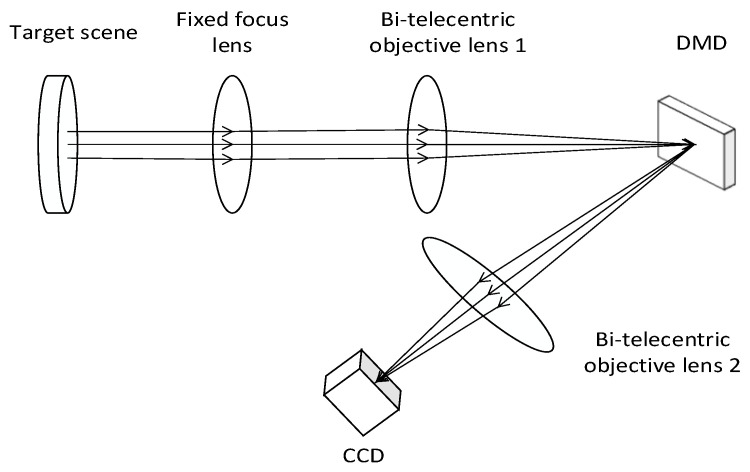
Experimental schematic diagram.

**Figure 12 sensors-22-02717-f012:**
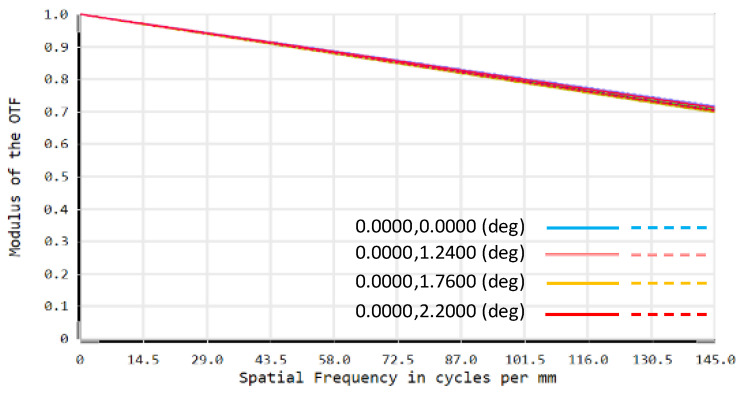
MTF of simulation system.

**Figure 13 sensors-22-02717-f013:**
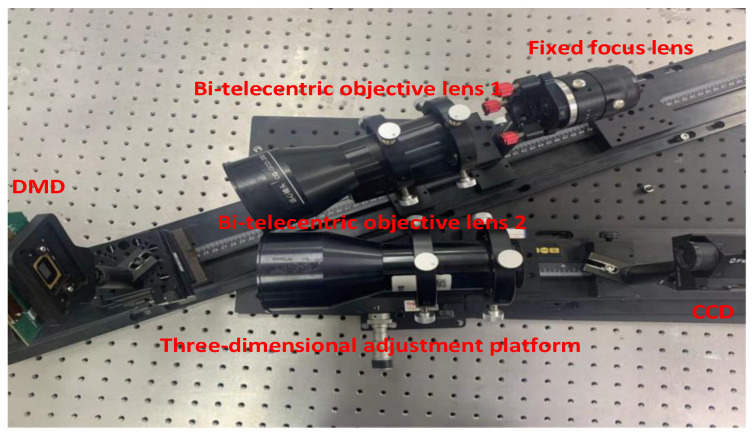
Experimental device.

**Figure 14 sensors-22-02717-f014:**
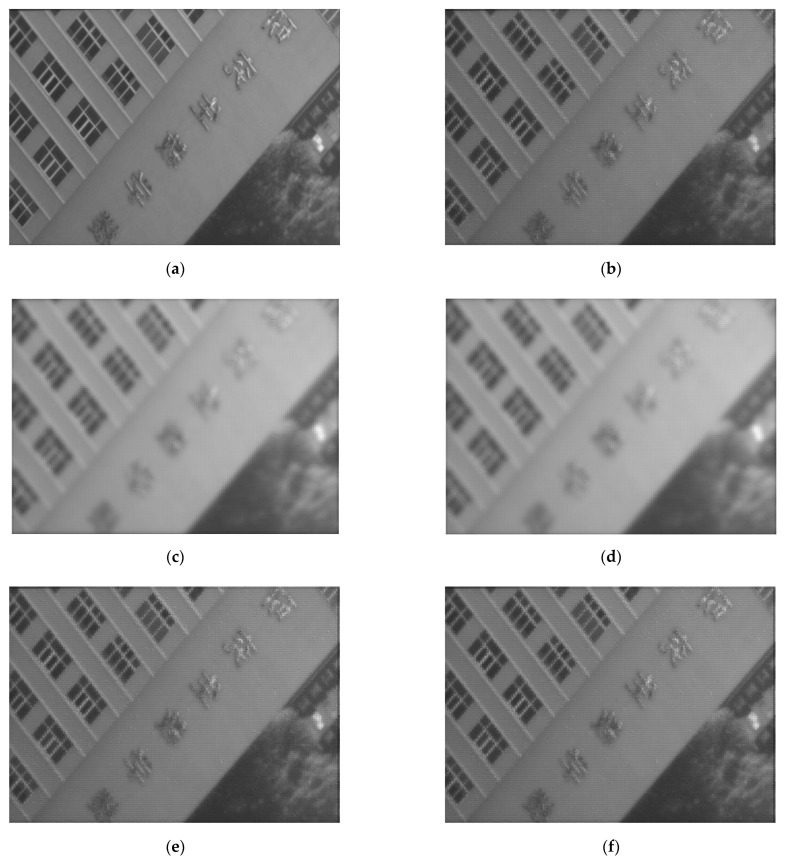
Experimental SR reconstruction images: (**a**) image with DMD fully lit and without pixel binning; (**b**) reconstructed image without error; (**c**) reconstruction image with 0.1 mm; (**d**) reconstruction image with 0.1 mm decenter in the X direction decenter in the Y direction; (**e**) reconstruction image with 0.1 mm; (**f**) reconstruction image with 0.1 mm tilt in the X direction tilt in the Y direction; and (**g**) reconstruction image with 0.1 mm defocus.

**Figure 15 sensors-22-02717-f015:**
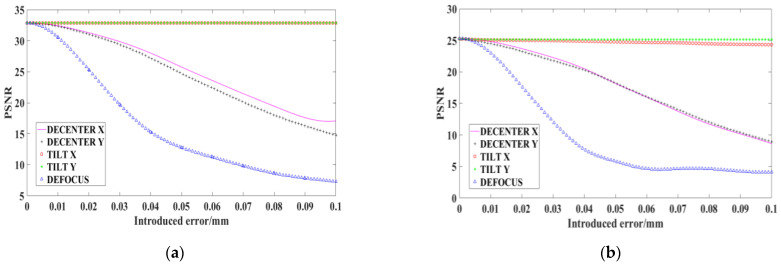
*PSNR* of the reconstructed image varies with the alignment error: (**a**) simulation; and (**b**) experiment.

**Table 1 sensors-22-02717-t001:** System parameters.

Parameter	Value
Wavelength/μm	8–12
FOV(X/Y)/(°)	4.4/3.52
F/#	1.76
DMD array size/pixel	1920 × 1080
DMD pixel size/μm	10.8
Detector pixel size/μm	17
Detector array size/pixel	480 × 270

**Table 2 sensors-22-02717-t002:** *PSNR* and SSIM of reconstructed images at various temperatures.

Temperature	*PSNR*	SSIM
−10 °C	22.3672	0.5037
10 °C	22.7532	0.5344
20 °C	22.9670	0.5509
40 °C	22.5427	0.5214

**Table 3 sensors-22-02717-t003:** Tolerace for alignment of the system.

Decenter (mm)		Tilt (°)		Lens Interval Error (mm)	Defocus of Intermediate Image Plane (mm)	Defocus of Image Plane (mm)
X	Y	X	Y	0.02	0.04	0.03
0.06	0.05	0.05	0.05			

## Data Availability

Data underlying the results presented in this Letter are not publicly available at this time but may be obtained from the authors upon reasonable request.

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
