# Peer review of "The Influence of Optical Alignment Error on Compression Coding Superresolution Imaging"

_sensors, 2022, doi:10.3390/s22072717_

Round 1

Reviewer 1 Report

Recommendation: Accept after revision

Comments:

The manuscript entitled “The Influence of Optical Alignment Error on Compression 1

Coding Superresolution Imaging.” proposed an algorithm strategy to compress images and maintain their super-resolution. Compressing images are critical for multi-functionalities but can reduce their resolution. In this manuscript, the authors did a great job for this task. I would like to recommend this paper can be published on Sensors after a minor revision.

I have some recommendations to help improve the quality of this paper:

  • In Fig.3, it is nice to see the schematic diagram of the optical system. I suggest they can merge this figure with Fig. 1. This would help the audience to better understand the method.
  • Some figures do not have clear words, such as Fig.5, 7, 13. Nobody can clearly identify those words.

Reviewer 2 Report

The authors use ZEMAX commercial software to analyze the effects of different alignment errors on super-resolution reconstructed image, and verified it by simulation and experiments. The compressed sensing super-resolution method has been proposed for a long time, such as [Stephan, H et al., IEEE TIM, 2013]. This manuscript is more inclined to engineering application analysis. Here are some suggestions to improve the work.

  1. As described in line 137, it uses 16 images with resolution of ab to reconstruct a high-resolution of 4a4b. Does that mean the compressed rate is 100%? I don't think this is a super-resolution work, but only exchanging temporal resolution for spatial resolution. Please give detailed super-resolution reconstruction steps.
  2. In the athermal analysis, only MTF curve was used to evaluate the performance of the optical system, and it was concluded that "the imaging quality of the system is well maintained in the operating temperature range." May the image reconstruction evaluation indexes such SSIM, PSNR or even the LPIPS loss should be given to support this conclusion.
  3. In the experimental part, the author used a CCD camera working in the visible band to verify the effect of the alignment error. However, the simulation results mentioned above is based on a long-wave infrared camera. I am not sure whether there is correlation between each other.
  4. The pixel-wised registration of DMD and camera is a difficult task, but the author used bin operation to avoid this process. It violates the original intention of super-resolution imaging task because the camera can directly collect the high-resolution images.
  5. The writing can be improved, for example, whether the magnification factor M is missed in formula 2; In line 92, ks,t is actually k_{s,t}; In line 126, the step(1) is not described before; In line 155, "MAXI is 2B-1" should be "MAX_{I}=2^{B}-1"; In line 253, it should be set to a tertiary tile . Please check the whole article carefully.
  6. The font of some pictures in the text is too small to see the labels, as shown in Fig.4,5,7,13. The resolution of figure 8,9,10,11,16 is too low, please meet the requirements of the journal.

Reviewer 3 Report

Dear authors, 

I have found this manuscript to be very nicely written, with a topic of interest to the readers of Sensor journal, and with very elaborately written method and results, which are followed by succinct and "straight to the point" conclusions. Except for some minor text editing, mainly in the introduction, and quality check of the figures, I think this Manuscript can be accepted for publication. Please find the annotated pdf attached with this letter, where you can find comments where some text or figure improvement is suggested. 

Best regards,

Round 2

Reviewer 2 Report

The authors have basically answered my question, but I still think this manuscript is more engineering application work, and not innovative enough